# Mortality and Years of Life Lost from Diabetes Mellitus in Poland: A Register-Based Study (2000–2022)

**DOI:** 10.3390/nu16213597

**Published:** 2024-10-23

**Authors:** Małgorzata Pikala, Monika Burzyńska

**Affiliations:** Department of Epidemiology and Biostatistics, The Chair of Social and Preventive Medicine, Medical University of Lodz, Żeligowskiego 7/9, 90-752 Lodz, Poland; malgorzata.pikala@umed.lodz.pl

**Keywords:** diabetes mellitus, T1DM, T2DM, mortality trends, standard expected years of life lost

## Abstract

Background: The aim of the study was to assess mortality and years of life lost from diabetes mellitus in Poland between 2000 and 2022. Methods: The database was created from death cards made available for the purposes of this study by Statistics Poland. The study used data on deaths caused by type 1 diabetes mellitus, T1DM (N = 33,328), and type 2 diabetes mellitus, T2DM (N = 113,706). Standardized death rates (SDRs) and standard expected years of life lost per person (SEYLL_p_) and per death (SEYLL_d_) were calculated. A time trend analysis was performed using joinpoint models. The annual percentage change (APC) and the average annual percentage change (AAPC) were estimated. Results: Between 2000 and 2022, 33,328 people died from T1DM in Poland. The SDR rate increased from 6.0 to 8.8 per 100,000 population in the analyzed period. The APC was 1.3% (*p* < 0.05). SEYLL_p_ rates per 100,000 population were 79.3 in 2000 and 109.2 in 2022. SEYLL_d_ rates were 22.9 and 17.9 years, respectively (APC = −1.0%, *p* < 0.05). The mean age of those who died from T1DM increased from 66.1 in 2000 to 72.5 in 2022. Between 2000 and 2022, 113,706 people died from T2DM. The SDR increased from 12.5 to 37.7 per 100,00 (APC = 5.5%, *p* < 0.05). SEYLL_p_ rates were 88.8 and 296.0 per 100,000 population (APC = 6.4%, *p* < 0.05). SEYLL_d_ rates decreased from 16.9 in 2000 to 13.4 in 2022 (AAPC = −1.0%, *p* < 0.05). The mean age of those who died from T2DM increased from 73.1 to 78.1 years. Conclusions: The study showed a growing problem of diabetes as a cause of death and years of life lost.

## 1. Introduction

Diabetes mellitus belongs to the group of civilization metabolic diseases. It develops when β-cells of the pancreas produce too little insulin or when tissues become insensitive to this hormone [1]. Type 1 and type 2 diabetes are most commonly diagnosed. Each has a different pathophysiology, but they both lead to similar complications [2]. Type 1 diabetes mellitus (T1DM) accounts for between 5% and 10% of all types of diabetes. It mainly affects children and people under 30 years of age. The onset of the disease is most often between 10 and 14 years of age. It is caused by the almost complete destruction of insulin-producing pancreatic β-cells by antibodies. These are so-called autoantibodies, produced by the body’s immune system, causing the destruction of its own cells. The phenomenon is called autoimmunity. It result is an absolute deficiency of insulin. Type 1 diabetes is not an inherited disease, but patients have a genetic predisposition to autoimmune diseases. It is believed that in genetically predisposed individuals, certain environmental factors can trigger an autoimmune response. The main genes associated with susceptibility to type 1 diabetes are HLA-DR3 and HLA-DR4 [3]. Type 2 diabetes mellitus (T2DM) accounts for approximately 90% of all diabetes cases. In T2DM, the response to insulin is impaired, which is referred to as insulin resistance. The development of the disease is determined by the presence of two primary abnormalities—decreased sensitivity of peripheral tissues to insulin and beta-cell secretory failure. The complex pathogenesis of diabetes, that is, the co-existence of insulin resistance and impaired insulin secretion by β-cells, makes the disease progressive [4]. T2DM most commonly occurs in people over 45 years of age. However, it is also increasingly being reported in children, adolescents, and young adults, which may be related to negative health behavior patterns. Indeed, the most important risk factors for T2DM, in addition to genetic factors and age, are excessive body weight, low physical activity, poor dietary habits, and comorbidities, including hypertension, non-alcoholic fatty liver disease (NAFLD), and gestational diabetes [5,6].

According to the World Health Organization (WHO) and the International Diabetes Federation (IDF), the number of people with diabetes has increased worldwide from approximately 108 million in 1980 to more than 460 million in 2019. By 2045, IDF projections show that one in eight adults, approximately 783 million, will be living with diabetes, an increase of 46% [7].

Accurate epidemiological data are essential for effective health care planning as they also hold significant prognostic value. However, the precise number of people with diabetes in Poland remains uncertain due to the absence of a comprehensive, population-based diabetes registry. Between 2006 and 2009, a pilot project for the Register of Adults with Diabetes in Poland was conducted. The aim of the project was to assess the quality of diabetes care. However, these registers are not fully operational today. Instead, the data used to estimate the scope of diabetes in Poland come from various studies conducted among different age groups, across different regions, or as part of research focused on other health issues, from which information about diabetes was extracted. The most frequently referenced studies, such as the regional Pol-MONICA and the nationwide multicenter WOBASZ II and NATPOL 2011 studies, primarily aimed to assess cardiovascular risk factors [8,9,10]. The POLSENIOR study evaluated the health status of individuals over 65 years old [11]. Meanwhile, data provided by WHO and IDF rely on information from individual countries, or, in the absence of such data, they are estimates based on information from countries with similar development levels and demographic profiles [12]. This lack of precise data has driven the search for alternative sources of reliable information to assess the prevalence of diabetes in Poland. In recent years, databases containing records of contracted medical services and reimbursed medications, maintained by the National Health Fund (NFZ), Poland’s sole public health insurer, have become valuable sources. Since 2014, following the introduction of Health Needs Maps in Poland, teams of experts and Ministry of Health officials have developed analyses to forecast and plan healthcare services, including those related to diabetes care.

Diabetes is a chronic disease that significantly impacts both the lifespan and quality of life of patients, often resulting in premature deaths. Analyzing mortality trends related to diabetes helps evaluate the success of preventive programs and therapeutic approaches. Understanding the epidemiological situation of diabetes also aids in better resource planning, such as ensuring the availability of specialized diabetes care, and highlights social groups that are at higher risk of mortality from this condition. Mortality data can inform the creation of strategies aimed at extending the lives of people with diabetes and enhancing their quality of life through improved disease management and complication prevention. This study is the first to focus on evaluating the health burden of diabetes in the Polish population, using comprehensive data on mortality and years of life lost due to diabetes over a span of more than 20 years.

According to the National Health Fund, in 2018, the number of new cases of diabetes of all types in Poland was 343,000. In 2022, there were almost 400,000 new diagnoses, and the total number of patients was 3.14 million [13]. The unfavorable epidemiological picture of incidence and prevalence translates into diabetes-related mortality. The aim of the study was to assess mortality and years of life lost from diabetes mellitus in Poland on the basis of the register-based study between 2000 and 2022.

## 2. Materials and Methods

The study used data on all deaths of Polish residents in the years 2000–2022 (N = 9,055,980). The database was created from death cards collected and made available for the purposes of this study by Statistics Poland. Deaths caused by type 1 diabetes mellitus (insulin-dependent diabetes, N = 33,328) and type 2 diabetes mellitus (noninsulin-dependent, N = 113,706) were isolated from the database. According to the International Statistical Classification of Diseases and Health Related Problems—Tenth Revision (ICD-10), these diseases were coded as E10 and E11, respectively. The procedure of coding causes of death in Poland is performed in a similar way to the one carried out in the majority of countries in the world, based on the so-called primary cause of death, or the disease that triggered a pathological process, leading to death. This is particularly important from the point of view of diseases that cause many complications, including diabetes. The quality of analyses conducted with the use of mortality statistical data depends on complete and reliable information included in death certificates but mainly on the proper and precise presentation of death causes. Poland is a country with 100% completeness of death registration, but the quality of cause of death coding was unsatisfactory. The World Health Organization tried to solve the problem of wrong coding of death causes by creating a list of so-called “garbage codes”, which should never be indicated as the primary death cause. Taking that into consideration, certain changes were introduced in Poland in 2009. In order to standardize death causes, which are subject to further statistical analyses, it was determined that the doctor who states the death is responsible for filing the death card, in which he or she puts the primary, secondary, and direct death causes, whereas qualified teams of doctors are responsible for coding death causes according to the ICD-10 classification. These duties of a dozen regional statistical offices were taken over by Statistics Poland.

In the analysis conducted for the purposes of this study, standardized death rates (SDRs) were calculated according to the following formula:$$SDR=\frac{\sum_{i=1}^{N}\frac{k_{i}}{p_{i}}w_{i}}{\sum_{i=1}^{N}w_{i}}$$
where *k_i_* is the number of diabetes deaths in this *i*-age group; *p_i_* is population size of this *i*-age group; *w_i_* is the weight assigned to this *i*-age group, resulting from the distribution of the standard population; and *N* is the number of age groups.

The standardization procedure was performed with the use of the direct method, in compliance with the European Standard Population, updated in 2013 [14]. In the direct standardization method, it is assumed that the age structure of the compared populations is identical to that in the standard population. Standardized mortality rates make it possible to determine the number of deaths that would occur in a given population if the age structure of this population were the same as the age structure of the population accepted as the standard, i.e., if the differences resulting from the different age structures of the compared populations were eliminated. The European Standard Population, revised in 2013, is the average population of EU and EFTA countries in each five-year age bracket.

Years of life lost were calculated and analyzed by the method described by Christopher Murray and Alan Lopez in Global Burden of Disease (GBD) [15]. The SEYLL index (standard expected years of life lost) was used to calculate the number of years of life lost by the studied population in comparison with the years lost by the referential (standard) population.

There are some methods of calculating lost years of life, and the main difference between them is a point of reference, i.e., the level of mortality which is considered “ideal”. In the GBD 2010 study, WHO experts recommend using life tables, based on the lowest noted death rate for each age group, in countries with populations above 5 million [16].

In this study, the SEYLL index was calculated according to the following formula:$$SEYLL=\sum_{x=0}^{l}d_{x}e_{x}^{*}$$
where $e_{x}^{*}$ is the life expectancy, based on GBD 2010 life tables; *d_x_* is the number of deaths at age *x*; *x* is the age at which the person died; and *l* is the last age that the population reaches.

The authors also applied the SEYLL per person (SEYLL_p_) index, which is a ratio of SEYLL and the size of the population (N), calculated per 100,000 population, and the SEYLL per death (SEYLL_d_) index, being a ratio of SEYLL and the number of deaths due to a particular cause, i.e., it expresses the number of YLL calculated per one dead person [17].
$$SEYLLp=\frac{\sum_{x=0}^{l}d_{x}e_{x}^{*}}{N}\times 100,000\;SEYLLd=\frac{\sum_{x=0}^{l}d_{x}e_{x}^{*}}{\sum_{x=0}^{l}d_{x}}$$

The 22-year trends in changes in SDR, SEYLL_p_, and SEYLL_d_ were determined for the years 2000–2022. The analysis of time trends was carried out with joinpoint models and the Joinpoint Regression Program (version 5.2.0), a statistical software package developed by the U.S. National Cancer Institute for the Surveillance, Epidemiology and End Results Program [18].

Time trends were determined with the use of segments joining in joinpoints, where trend values significantly changed (*p* < 0.05) [19]. To confirm whether the changes were statistically significant, the Monte Carlo Permutation method was applied.

The annual percentage change (APC) in standardized mortality rates was estimated for each section of broken straight lines, as well as the average annual percentage rates of change (AAPC is the average annual percentage change) for the entire analysis period, along with the corresponding 95% confidence intervals (CIs).

APC coefficients were calculated according to the following formula:$$APC=100\times ({exp}^{b}-1)$$
where b is the slope coefficient.

This approach assumes that death rates change at a constant percentage from the previous year’s rate. For example, if the APC is 1% and the SDR in 2000 is 50 per 100,000 population, the SDR in 2001 is 50 × 1.01 = 50.5; in 2002, 50.5 × 1.01 = 51,005, etc.

The average annual percentage change (AAPC) is a summary measure of the trend over a predefined fixed period of time. It allows you to use a single number to describe the average APC over a period of many years. This is important when the joinpoint model indicates that there have been changes in trends in these years. AAPC is calculated as the weighted average of the APC from the joinpoint model, with weights equal to the length of the APC interval [20].
$$AAPC=\left\{exp\left(\frac{\sum w_{i}b_{i}}{\sum w_{i}}\right)-1\right\}\times 100$$
where $b_{i}$ is the slope coefficient for each interval in the analyzed period and $w_{i}$ is the length of each interval in the analyzed period.

## 3. Results

Between 2000 and 2022, 147,034 people died from diabetes mellitus in Poland, including 33,328 from type 1 diabetes (T1DM) and 113,706 from type 2 diabetes (T2DM) (Table 1 and Table 2).

For deaths due to T1DM, the standardized death rate (SDR) was 6.0 per 100,000 population in 2000 and increased to 8.8 per 100,000 population in 2022 (Table 1). The annual percentage change (APC) between 2000 and 2022 was 1.3% (*p* < 0.05) (Table 3, Figure 1a). Higher SDR values and a faster growth rate were observed for the male group. The SDR due to T1DM in the male group increased from a value of 6.2 in 2000 to 10.3 in 2022 (APC = 2.1%, *p* < 0.05). Among females, the SDR was 5.7 in 2000 and 7.6 in 2022. The trend changes were statistically insignificant (APC = 0.7%, *p* > 0.05) (Table 1 and Table 3, Figure 1a).

Considerably higher death rates were associated with type 2 diabetes mellitus. The SDR increased from 12.5 in 2000 to 37.7 in 2022 (APC = 5.5%, *p* < 0.05) (Table 2 and Table 4, Figure 1b). In the male group, the SDR was 12.1 in 2000 and 40.5 in 2022 (APC = 6.3%, *p* < 0.05). In the female group, the SDR values were 12.6 and 35.2 in the first and last years of the analyzed period, respectively (APC = 5.1%, *p* < 0.05) (Table 2 and Table 4, Figure 1b).

The number of standard expected years of life lost (SEYLL) due to deaths caused by T1DM was 30,352 in 2000 and 41,224 in 2022 (Table 1). The values of SEYLL_p_ rates per 100,000 population in Poland were 79.3 and 109.2, respectively. During the analyzed period, the trends’ direction and rate changed twice. A statistically insignificant decrease in SEYLL_p_ rates between 2000 and 2002 was followed by an increased rate of 1.3% per year (*p* < 0.05) between 2002 and 2018 and then a more rapid increase of 7.9% per year (*p* < 0.05) between 2018 and 2022. The average annual percentage change (AAPC) throughout the analyzed period was statistically insignificant (Table 3, Figure 1c).

Among males, the number of life years lost due to T1DM-related deaths was 16,605 in 2000 and increased to 25,024 in 2022. The SEYLL_p_ per 100,000 males increased from 89.6 to 137.1 (Table 1). After non-significant statistical changes in the rate between 2000 and 2015, the SEYLL_p_ increased at a rate of 5.7% (*p* < 0.05) between 2015 and 2022. The AAPC was 2.5% (*p* < 0.05) throughout the analysis period (Table 3, Figure 1c).

Among females, deaths due to T1DM accounted for 13,748 years of life lost in 2000 and 16,200 years in 2022. The SEYLL_p_ rate per 100,000 females increased from 69.7 to 83.0 (Table 1). A statistically insignificant decline in SEYLL_p_ values between 2000 and 2004 was followed by an increase at a rate of 1.6% per year (*p* < 0.05). Changes over the whole analyzed period were statistically insignificant (Table 3, Figure 1c).

For deaths due to type 2 diabetes mellitus, the number of life years lost was 33,952 in 2000 and increased to 111,803 years in 2022. The SEYLL_p_ rates were 88.8 and 296.0 per 100,000 population (APC = 6.4%, *p* < 0.05) (Table 2 and Table 4, Figure 1d). In the male group, the number of life years lost increased from 14,558 in 2000 to 60,881 in 2022. Per 100,000 males, the SEYLL_p_ values were 78.5 and 333.6 (APC = 7.5%, *p* < 0.05).

Among females, the number of life years lost increased from 19,393 in 2000 to 50,922 in 2022, and SEYLL_p_ rates increased from 98.4 to 260.9 (APC = 5.3%, *p* < 0.05) (Table 2 and Table 4, Figure 1d).

SEYLL_d_ rates representing the number of life years lost per person who died from a given cause were also calculated. For type 1 diabetes mellitus, each person who died from the disease lost on average 22.9 years in 2000, and 17.9 years in 2022 (APC = −1.0%, *p* < 0.05) (Table 1 and Table 3, Figure 1e). The decrease in SEYLL_d_ values was directly related to the increase in the median age of those who died from T1DM. In 2000, the median age was 69 years (25–75%: 58–76), increasing to 74 years (25–75%: 65–83) in 2022 (Figure 2a).

Among males, SEYLL_d_ rates due to T1DM decreased from 26.7 in 2000 to 21.2 in 2022 (APC = −1.0%, *p* < 0.05) (Table 1 and Table 3, Figure 1e). The median age of males who died from T1DM was 64 years (25–75%: 51–72) in 2000 and 70 years (25–75%: 60–77) in 2022 (Figure 2b).

**Figure 1 nutrients-16-03597-f001:**
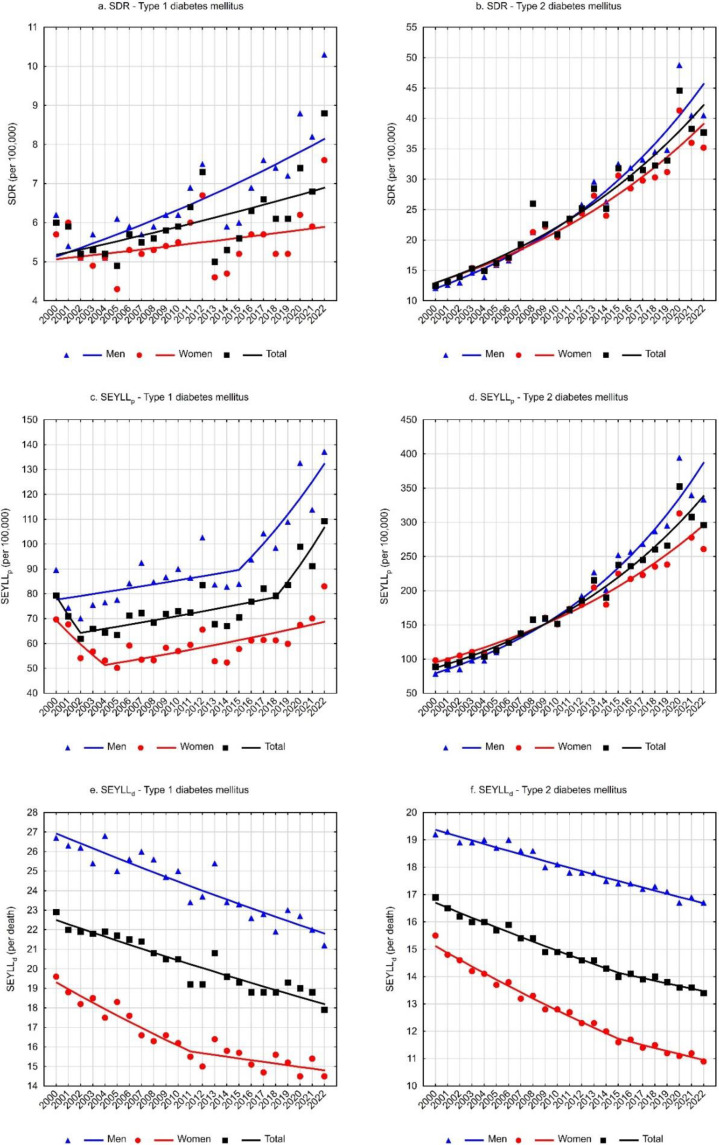
Time trends of standardized death rate (SDR), standard expected years of life lost (SEYLL), standard expected years of life lost per person (SEYLL_p_), and standard expected years of life lost per death (SEYLL_d_) due to diabetes mellitus in Poland in 2000–2022.

In the female group, deaths due to T1DM contributed to an average loss of 19.6 years of life for each female dying from this cause in 2000 and 14.5 years in 2022 (AAPC = −1.2%, *p* < 0.05). The rate of decline in SEYLL_d_ decreased from −1.8% (*p* < 0.05) between 2000 and 2011 to −0.6% (*p* < 0.05) between 2011 and 2022 (Table 1 and Table 3, Figure 1e). The median age of females who died from T1DM increased from 72 years in 2000 (25–75%: 65–77) to 79 years in 2022 (25–75%: 70–87) (Figure 2c).

For type 2 diabetes mellitus, SEYLL_d_ rates decreased from 16.9 in 2000 to 13.4 in 2022 (AAPC = −1.0%, *p* < 0.05). The rate of decline between 2000 and 2015 was −1.1% (*p* < 0.05) and decreased between 2015 and 2022 to −0.7% (*p* < 0.05) (Table 2 and Table 4, Figure 1f). The median age of those who died from T2DM was 74 years in 2000 (25–75%: 67–80), increasing to 80 years in 2022 (25–75%: 71–87) (Figure 2d).

Among males, SEYLL_d_ rates due to T2DM were 19.2 years in 2000 and 16.7 years in 2022 (APC = −0.7%, *p* < 0.05). The median age of males who died from this cause increased from 71 years in 2000 (25–75%: 65–78) to 74 years in 2022 (25–75%: 66–83) (Figure 2e).

Among females, T2DM contributed to an average loss of 15.5 years of life in each female who died from this cause in 2000 and 10.9 years of life in 2022. The rate of decline in SEYLL_d_ rates was −1.7% (*p* < 0.05) between 2000 and 2015 and −1.0% (*p* < 0.05) between 2015 and 2022. The AAPC for the entire analyzed period was −1.5% (*p* < 0.05) (Table 2 and Table 4, Figure 1e). The median age of females who died from T2DM increased from 75 years in 2000 (25–75%: 69–81) to 84 years in 2022 (25–75%: 75–89) (Figure 2f).

**Figure 2 nutrients-16-03597-f002:**
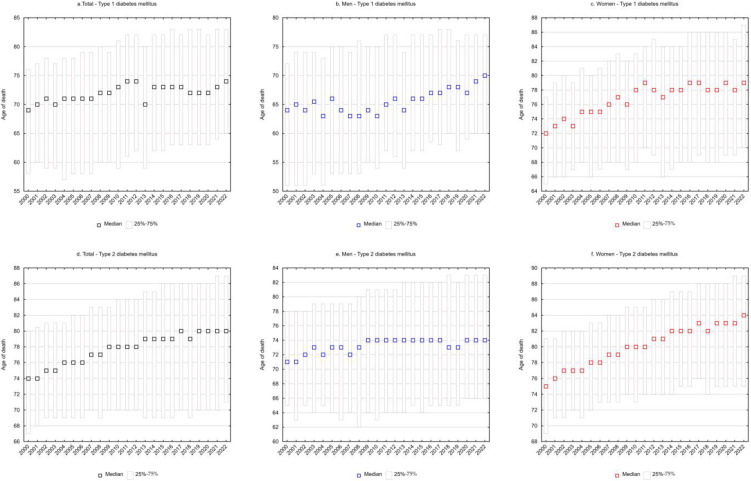
Median age of persons who died from diabetes mellitus in Poland in 2000–2022.

## 4. Discussion

Diabetes has been a growing medical, social, and economic problem for years. It ranks among the top 10 causes of disability for people worldwide [21]. Of four million people who die each year from this cause, a significant proportion, i.e., between 6% and 27%, are economically active (35–60 years) [7]. The average global trend in diabetes incidence from 1990 to 2019 was upward, with an annual increase of 3.73 cases per 100,000 population. The global average diabetes mortality rate showed an upward trend with an annual increase of 0.43 cases until 2005, followed by a downward trend after that year with an annual decrease of 0.14, and the global mortality-to-incidence ratio showed over the same period a downward trend with an annual decline of 0.001 per 100,000 population. Most people with diabetes (three in four people) live in low- and middle-income countries. The largest number of patients is recorded in the Western Pacific area (206 million). This is followed by people in Southeast Asia (90 million), the Middle East and North Africa (73 million), Europe (61 million), North America and the Caribbean (51 million), South and Central America (32 million), and Africa (24 million) [22]. More than one in two (54%) people with diabetes remain undiagnosed [7]. The mortality situation due to diabetes, both type 1 and type 2, in Poland is worse than in other European countries. In 2021, the highest mortality rate due to this cause was recorded in the Czech Republic (43.4 per 100,000 population) and the lowest in Switzerland (9.6 per 100,000 population). In Poland, it amounted to 29.9 per 100,000 population and was the fifth highest on the continent. At the global level, on the other hand, it was on average 19.6 per 100,000 population. Data on mortality in people with type 1 diabetes aged 0–79 years from Australia, Denmark, Latvia, Scotland, Spain, and the United States showed a decrease in age- and sex-standardized all-cause mortality rates in people with type 1 diabetes with annual estimated changes in mortality rates ranging from −2.1% to −5.8%. Mortality from T2DM has generally decreased in EU countries also between 1990 and 2019. Over the 30 years included in the analysis, a decrease in the age-standardized mortality rate (ASMR) was observed in 16 of 28 countries for males and in 24 of 28 countries for females. The highest relative decline in the rate was observed for males in the UK (−46.9%). For females, the highest relative decrease in ASMR was in Cyprus (−67.6%) [23,24]. Our study showed a worse epidemiological situation of diabetes in males than in females. Simultaneously, recent studies have found a higher prevalence of type 2 diabetes in males than in females. Males are usually diagnosed at a younger age [25,26,27] also, but the reason for this difference has been unclear. It is usually attributed to more difficult glycemic control, biological differences, behavioral factors, and lifestyles. Males are also at greater risk of the complications associated with diabetes [28].

The trends in diabetes observed in Poland were not favorable. In fact, our study showed an increase in standardized mortality rates for both types of diabetes between 2000 and 2022. Similar results were observed in other studies. Hospital mortality due to diabetes rose up dangerously to 3.77% between 2010 and 2018 [29]. During the first year of the COVID-19 pandemic in Poland, a markable decrease in diabetes-related hospitalization rates and an increase in in-hospital mortality rates have been observed [30]. These trends are consistent with incidence trends [31]. In the last decade, there has been an increase in the incidence of type 2 diabetes, which, among other things, is attributed to the development of diagnostics of diabetes and improved detection of the disease in the general population and increasingly longer life expectancy, especially in females. Epidemiological data also indicate that age and modified lifestyle factors are the strongest risk factor for diabetes (diabetes is diagnosed in every fourth person over 60 years of age) [32]. An increasing trend in the incidence is also observed for type 1 diabetes. At the end of the 20th century, the incidence rate in Poland was 15 cases per 100,000 population, while at present it is about 25 per 100,000 population. Currently, the total number of patients is estimated at nearly 180,000, including approximately 20,000 recorded cases of the disease among pediatric and adolescent populations [33]. As a consequence, a statistically significant increase in the number of years of life lost (SEYLL_p_) has been observed since 2004 in the female group and since 2015 in the group of males.

Studies suggest that the body mass index (BMI) may be a determining factor when comparing the two diabetes subtypes. An elevated BMI is more common in patients diagnosed with type 2 diabetes [34,35]. The pandemic, closely connected with social isolation, contributed to the occurrence of negative lifestyles, such as sedentary lifestyles and increased calorie intake. They were exacerbated and contributed to an increased body mass index in the population. According to the Centers for Disease Control and Prevention (CDC), BMI doubled during the pandemic compared with the pre-pandemic period [36].

Epidemiological studies show that more than 85% of patients with type 2 diabetes are overweight or obese. The contribution of high BMI to disability adjusted life years (DALYs) in type 2 diabetes increased by 24.3% between 1990 and 2021 [23]. However, the impact of BMI on mortality among patients with diabetes is still controversial. Recently, the phenomenon of the “obesity paradox”, which refers to a lower risk of death in overweight or obese patients assessed by BMI, has been reported in various populations [37]. A meta-analysis of 20 studies involving 250,016 patients showed a lower risk of death in overweight/obese patients with coexisting diabetes compared with normal-weight patients, whereas the beneficial prognostic impact of obesity was more pronounced in older adult patients but weakened with longer follow-up [38]. However, the results of the study should be carefully interpreted as the design of the study may not allow for verification of the causal relationships between body weight and prognosis in patients with diabetes, and the results only confirm statistical correlations rather than a causal relationship [39]. At the same time, the deleterious effects of obesity on metabolism and insulin resistance are well documented in the literature, and obesity has been shown to be closely linked to the etiology of type 2 diabetes and is one of the most important risk factors for the development of the disease [40].

In Poland, the proportion of overweight and obese people has been increasing over the years. According to the Global Obesity Observatory, the proportion of overweight and obese people in Poland increased from 40% to 50% between 2000 and 2020 [41]. The results of a study on the prevalence of general and abdominal obesity and overweight in adult Poles in 2017–2020, involving 2000 randomly selected residents, revealed that excessive body weight was found in 51% of respondents (55% males and 47% females). Abdominal overweight was found in 21.2% and abdominal obesity in 27.2% of respondents. At the same time, visceral fat distribution is a major risk factor for metabolic diseases, including type 2 diabetes, which may indirectly translate into increased morbidity and mortality rates from this cause. Males were 43% more likely to be overweight than females, and the risk increased with age [42]. This disparity in overweight/obesity risk factor exposure in relation to sex corresponds with the results of our study, in which higher SDRs and a faster rate of increase were specifically in the male population.

Low physical activity composes another risk factor for diabetes. Inactivity slows down glucose transport and oxidation. In contrast, regular physical activity (PA) reduces the risk of type 2 diabetes by up to 60% [43], as well as enhances health and glycemic management in people with already diagnosed disease [44]. A cohort study involving 19,624 people with type 2 diabetes mellitus (T2DM) registered in a UK biobank, with a mean follow-up of 6.9 years, found that physical activity of any intensity was associated with a lower risk of mortality among patients with T2DM compared with those who were physically inactive [45]. The European Prospective Investigation into Cancer and Nutrition (EPIC) study, involving 5859 people with diabetes, confirmed that the risk of premature death in patients who were engaged in moderate activity was significantly lower compared with those not physically active [46]. A beneficial effect of physical activity has also been observed in relation to T1DM. In these cases, it can lead to improved insulin sensitivity, improved muscle strength, and cardiovascular fitness [47].

Data on physical activity in EU countries, gathered by the Organization for Economic Co-operation and Development, show that physical activity levels changed significantly between 2017 and 2022. In five EU countries, the percentage of people claiming that they never play or are engaged in sports has decreased significantly: Malta (−25%, now 31%), Latvia (−23%, 33%), Estonia (−18%, 30%), Croatia (−16%, 40%), and the Czech Republic (−15%, 26%). Unfortunately, Poland is one of the countries where this percentage has increased most (+9.5%), along with Hungary (+6.59%) and Portugal (+5.73%). Furthermore, Poland is one of eight countries with the highest percentage of people who never exercise or play sports [48]. This is another factor whose prevalence may be related to the observed adverse trends in diabetes mortality in the Polish population.

In developing countries, poor dietary habits are also one of the main factors contributing to the rapidly increasing incidence of diabetes. In turn, elevated HbA1c levels, which can be improved by an adequate diet, have recently been recognized as a risk factor for microvascular and macrovascular complications [49]. Diets rich in whole grains, fruits, vegetables, legumes, and nuts, with moderate alcohol consumption, fewer refined grains, less red/processed meat, and fewer sugar-sweetened beverages have been shown to reduce the risk of diabetes and improve glycemic control and blood lipid levels in patients with diabetes [50]. During the COVID-19 pandemic, numerous negative changes in the diet and lifestyle of Poles were observed [51]. The quality of dietary habits of the majority of the Polish adult population deviates from recommendations [52], which may contribute to an increased prevalence of T2DM in the Polish population.

It is worth noting the large increase in deaths from both types of diabetes observed in our study with the onset of the COVID-19 pandemic. During SARS-CoV-2 infection, already diagnosed diabetes may in turn become clinically and therapeutically unstable. Insulin requirements then increase, which can lead to the development of ketoacidosis. In combination with risk factors such as obesity and glucose intolerance, this results in a more severe course of infection and increases the risk of death [53]. This explains the increase in mortality from diabetes observed in our study. Similar results have also been confirmed in other studies. Patients with diabetes are more prone to lower respiratory tract infections, mainly due to impaired immune function, including a lack of normal phagocytosis by neutrophils, macrophages, and monocytes. Hyperglycemia, in turn, is associated with an increased risk of severe infections [54]. Therefore, patients with both type 1 and type 2 diabetes demonstrate a higher risk of developing more severe forms of COVID-19 and even death [55]. In patients with co-morbidities, irrespective of diabetes subtype, mortality is associated with age, male sex, concomitant diseases (primarily cardiovascular disease—CVD, kidney disease), obesity, and being underweight [56].

Life years lost due to both diabetes subtypes were also analyzed in our study. The number of life years lost for both T1DM and T2DM in both sex groups significantly increased, which correlated with adverse mortality trends. However, in contrast to the observed unfavorable changes in the number of life years lost per 100,000 population (SEYLL_p_), the results of our analysis indicated concurrent favorable trends in life years lost due to diabetes of both subtypes per person who died from this cause (SEYLL_d_). The average age of dying people also increased. This was closely linked to improved diagnosis and treatment efficacy, which, in turn, improved survival rates. Modern drugs and glycemic monitoring systems have revolutionized the treatment of diabetes all around the world [57]. Also in Poland, due to successive financial investments, modern diabetes treatment and monitoring technologies became available. Other studies also confirmed improved survival rates. A multi-cohort study of more than 40,000 people with type 1 diabetes in Finland revealed that survival improved significantly between 1972 and 2017 [58]. These results proved the improved efficacy of T1DM treatment and improved control of comorbidities. It has also been shown that the life expectancy of people diagnosed in recent decades increased on average by 15 years in comparison with those diagnosed in the 1960s or earlier [59]. Improved survival rates have also been reported for type 2 diabetes if the therapy was controlled and the patient adhered to recommendations [60]. A study based on data from 23 countries around the world confirmed both favorable trends in SEYLL_d_ rates (the fastest decline was observed in males in the United States) and the increased life expectancy in people with type 2 diabetes. Life expectancy for these people increased both in males and females between 2005 and 2019 in all analyzed countries except Spain and Scotland [61]. In Poland, during the analysis period, the average age of people dying from type 1 diabetes increased by more than 6 years and type 2 diabetes by more than 5 years.

Epidemiological studies indicate that diabetes is a serious public health problem. This has been contributed by population ageing, as well as increased prevalence of overweight and obesity, associated with poor nutrition and sedentary lifestyles.

## 5. Conclusions

The analysis of diabetes mortality trends in Poland during the analyzed period indicates a growing problem of diabetes, particularly type 2, as a cause of death and years of life lost. Although the increase in the average age at death suggests improvements in the quality of medical care and better disease control, the number of premature deaths related to diabetes continues to rise, especially among males. The increase in the number of years of life lost and the number of deaths highlights the need to intensify preventive measures, improve diabetes management, and provide more effective support for patients in controlling the disease.

## Figures and Tables

**Table 1 nutrients-16-03597-t001:** Number of deaths and values of standardized death rate (SDR), standard expected years of life lost (SEYLL), standard expected years of life lost per person (SEYLL_p_), and standard expected years of life lost per death (SEYLL_d_) due to type 1 diabetes mellitus in Poland in 2000–2022.

Year	Men					Women					Total				
	Number of Deaths	SDR (per 100,000)	SEYLL	SEYLL_p_ (per 100,000)	SEYLL_d_ (per Deaths)	Number of Deaths	SDR (per 100,000)	SEYLL	SEYLL_p_ (per 100,000)	SEYLL_d_ (per Deaths)	Number of Deaths	SDR (per 100,000)	SEYLL	SEYLL_p_ (per 100,000)	SEYLL_d_ (per Deaths)
2000	621	6.2	16,605	89.6	26.7	703	5.7	13,748	69.7	19.6	1324	6.0	30,352	79.3	22.9
2001	524	5.4	13,792	74.4	26.3	712	6.0	13,357	67.7	18.8	1236	5.9	27,149	71.0	22.0
2002	496	5.1	12,976	70.1	26.2	587	5.1	10,688	54.2	18.2	1083	5.2	23,664	61.9	21.9
2003	550	5.7	13,961	75.5	25.4	606	4.9	11,197	56.8	18.5	1156	5.3	25,158	65.9	21.8
2004	527	5.1	14,128	76.5	26.8	599	5.1	10,490	53.2	17.5	1126	5.2	24,618	64.5	21.9
2005	573	6.1	14,322	77.6	25.0	541	4.3	9,900	50.2	18.3	1114	4.9	24,222	63.5	21.7
2006	605	5.9	15,513	84.2	25.6	661	5.3	11,660	59.2	17.6	1266	5.7	27,173	71.3	21.5
2007	655	5.7	17,023	92.5	26.0	636	5.2	10,545	53.5	16.6	1291	5.5	27,568	72.3	21.4
2008	610	5.9	15,598	84.7	25.6	644	5.3	10,516	53.3	16.3	1254	5.6	26,113	68.5	20.8
2009	648	6.2	15,980	86.7	24.7	693	5.4	11,514	58.3	16.6	1341	5.8	27,494	72.0	20.5
2010	671	6.2	16,788	90.0	25.0	699	5.5	11,333	57.0	16.2	1370	5.9	28,121	73.0	20.5
2011	688	6.9	16,116	86.4	23.4	764	6.0	11,826	59.5	15.5	1452	6.4	27,941	72.5	19.2
2012	806	7.5	19,141	102.6	23.7	869	6.7	13,042	65.6	15.0	1675	7.3	32,183	83.5	19.2
2013	616	5.0	15,599	83.7	25.4	639	4.6	10,505	52.9	16.4	1255	5.0	26,103	67.8	20.8
2014	658	5.9	15,416	82.8	23.4	657	4.7	10,404	52.4	15.8	1315	5.3	25,819	67.1	19.6
2015	671	6.0	15,613	83.9	23.3	730	5.2	11,471	57.8	15.7	1401	5.6	27,084	70.5	19.3
2016	772	6.9	17,438	93.8	22.6	803	5.7	12,134	61.2	15.1	1575	6.3	29,572	76.9	18.8
2017	851	7.6	19,387	104.3	22.8	826	5.7	12,176	61.4	14.7	1677	6.6	31,563	82.1	18.8
2018	835	7.4	18,277	98.4	21.9	780	5.2	12,157	61.3	15.6	1615	6.1	30,434	79.2	18.8
2019	880	7.2	20,219	108.9	23.0	781	5.2	11,865	59.9	15.2	1661	6.1	32,084	83.6	19.3
2020	1077	8.8	24,539	132.6	22.7	920	6.2	13,344	67.5	14.5	1997	7.4	37,883	99.0	19.0
2021	950	8.2	20,940	113.8	22.0	896	5.9	13,797	70.1	15.4	1846	6.8	34,737	91.2	18.8
2022	1178	10.3	25,024	137.1	21.2	1120	7.6	16,200	83.0	14.5	2298	8.8	41,224	109.2	17.9
**Total**	**16,462**					**16,866**					**33,328**				

**Table 2 nutrients-16-03597-t002:** Number of deaths and values of standardized death rate (SDR), standard expected years of life lost (SEYLL), standard expected years of life lost per person (SEYLL_p_), and standard expected years of life lost per death (SEYLL_d_) due to type 2 diabetes mellitus in Poland in 2000–2022.

Year	Men					Women					Total				
	Number of Deaths	SDR (per 100,000)	SEYLL	SEYLL_p_ (per 100,000)	SEYLL_d_ (per Deaths)	Number of Deaths	SDR (per 100,000)	SEYLL	SEYLL_p_ (per 100,000)	SEYLL_d_ (per Deaths)	Number of Deaths	SDR (per 100,000)	SEYLL	SEYLL_p_ (per 100,000)	SEYLL_d_ (per Deaths)
2000	757	12.1	14,558	78.5	19.2	1251	12.6	19,393	98.4	15.5	2008	12.5	33,952	88.8	16.9
2001	816	12.6	15,780	85.2	19.3	1308	13.2	19,366	98.2	14.8	2124	13.1	35,145	91.9	16.5
2002	835	13.0	15,756	85.1	18.9	1417	14.1	20,715	105.1	14.6	2252	13.9	36,470	95.4	16.2
2003	955	14.6	18,061	97.7	18.9	1536	15.4	21,801	110.6	14.2	2491	15.3	39,862	104.4	16.0
2004	951	13.9	18,025	97.6	19.0	1527	15.0	21,573	109.5	14.1	2478	14.9	39,598	103.7	16.0
2005	1085	15.9	20,252	109.7	18.7	1652	16.0	22,660	115.0	13.7	2737	16.2	42,912	112.5	15.7
2006	1200	16.6	22,844	124.0	19.0	1775	16.9	24,430	124.0	13.8	2975	17.1	47,274	124.0	15.9
2007	1372	18.8	25,545	138.7	18.6	2026	19.0	26,812	136.1	13.2	3398	19.3	52,357	137.4	15.4
2008	1558	21.3	28,983	157.4	18.6	2346	21.3	31,177	158.1	13.3	3904	26.0	60,161	157.8	15.4
2009	1628	22.6	29,247	158.7	18.0	2469	22.2	31,671	160.5	12.8	4097	22.6	60,917	159.6	14.9
2010	1571	20.9	28,469	152.6	18.1	2339	20.5	29,976	150.8	12.8	3910	20.9	58,445	151.7	14.9
2011	1793	23.6	31,920	171.1	17.8	2707	23.0	34,483	173.4	12.7	4500	23.5	66,402	172.3	14.8
2012	2010	25.8	35,851	192.2	17.8	2899	24.4	35,684	179.5	12.3	4909	25.2	71,535	185.6	14.6
2013	2369	29.6	42,254	226.8	17.8	3314	27.3	40,663	204.7	12.3	5683	28.5	82,918	215.4	14.6
2014	2137	26.3	37,454	201.2	17.5	2962	24.0	35,682	179.7	12.0	5099	25.2	73,136	190.1	14.3
2015	2702	32.5	46,886	252.1	17.4	3837	30.6	44,598	224.8	11.6	6539	31.8	91,484	238.0	14.0
2016	2745	31.9	47,708	256.6	17.4	3676	28.5	43,098	217.2	11.7	6421	30.2	90,806	236.3	14.1
2017	2904	33.2	49,945	268.6	17.2	3887	29.8	44,218	222.9	11.4	6791	31.5	94,163	245.0	13.9
2018	3088	34.5	53,298	286.8	17.3	4046	30.3	46,637	235.2	11.5	7134	32.3	99,936	260.2	14.0
2019	3209	34.8	54,845	295.4	17.1	4200	31.2	47,226	238.3	11.2	7409	33.1	102,071	265.9	13.8
2020	4368	48.8	73,007	394.6	16.7	5557	41.3	61,868	313.1	11.1	9925	44.6	134,875	352.5	13.6
2021	3712	40.5	62,605	340.1	16.9	4887	36.0	54,639	277.7	11.2	8599	38.3	117,244	307.9	13.6
2022	3643	40.5	60,881	333.6	16.7	4680	35.2	50,922	260.9	10.9	8323	37.7	111,803	296.0	13.4
**Total**	**47,408**					**66,298**					**113,706**				

**Table 3 nutrients-16-03597-t003:** Time trends of standardized death rate (SDR), standard expected years of life lost per person (SEYLL_p_), and standard expected years of life lost per death (SEYLL_d_) due to type 1 diabetes mellitus in Poland in 2000–2022—joinpoint regression analysis.

	Number of Joinpoints	Years	APC (95% CI)	AAPC (95% CI)
Males				
SDR	0	2000–2022	2.1 * (1.3; 2.9)	
SEYLL_p_	1	2000–2015	1.0 (−0.1; 2.0)	2.5 * (1.2; 3.7)
		2015–2022	5.7 * (2.3; 9.3)	
SEYLL_d_	0	2000–2022	−1.0 * (−1.1; −0.8)	
Females				
SDR	0	2000–2022	0.7 (−0.1; 1.5)	
SEYLL_p_	1	2000–2004	−7.3 (−14.4; 0.4)	0.0 (−1.5; 1.4)
		2004–2022	1.6 * (0.8; 2.5)	
SEYLL_d_	1	2000–2011	−1.8 * (−2.4; −1.3)	−1.2 * (−1.6; −0.8)
		2011–2022	−0.6 * (−1.1; −0.1)	
Total				
SDR	0	2000–2022	1.3 * (0.6; 2.0)	
SEYLL_p_	2	2000–2002	−0.9 (−26.5; 10.5)	1.4 (−0.7; 3.5)
		2002–2018	1.3 * (0.4; 2.1)	
		2018–2022	7.9 * (1.2; 15.1)	
SEYLL_d_	0	2000–2022	−1.0 * (−1.1; −0.8)	

* *p* < 0.05.

**Table 4 nutrients-16-03597-t004:** Time trends of standardized death rate (SDR), standard expected years of life lost per person (SEYLL_p_), and standard expected years of life lost per death (SEYLL_d_) due to type 2 diabetes mellitus in Poland in 2000–2022—joinpoint regression analysis.

	Number of Joinpoints	Years	APC (95% CI)	AAPC (95% CI)
Males				
SDR	0	2000–2022	6.3 * (5.7; 6.8)	
SEYLL_p_	0	2000–2022	7.5 * (7.0; 8.0)	
SEYLL_d_	0	2000–2022	−0.7 * (−0.7; −0.6)	
Females				
SDR	0	2000–2022	5.1 * (4.7; 5.6)	
SEYLL_p_	0	2000–2022	5.3 * (4.8; 5.8)	
SEYLL_d_	1	2000–2015	−1.7 * (−1.8; −1.5)	−1.5 * (−1.6; −1.3)
		2015–2022	−1.0 * (−1.4; −0.5)	
Total				
SDR	0	2000–2022	5.5 * (4.9; 6.1)	
SEYLL_p_	0	2000–2022	6.4 * (5.9; 6.9)	
SEYLL_d_	1	2000–2015	−1.1 * (−1.2; −1.0)	−1.0 * (−1.1; −0.9)
		2015–2022	−0.7 * (−1.0; −0.4)	

* *p* < 0.05.

## Data Availability

The data presented in this study are available on request from the corresponding author due to privacy.

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
