# Peer review of "Mortality and Years of Life Lost from Diabetes Mellitus in Poland: A Register-Based Study (2000–2022)"

_nutrients, 2024, doi:10.3390/nu16213597_

Round 1
Reviewer 1 Report
Comments and Suggestions for Authors
General comments.
The aim of the study is formulated as follows: “The aim of the study was to assess mortality and years of life lost from diabetes mellitus in Poland on a base of the register-based study between 2000 and 2022.” This assessment can certainly add to the knowledge and understanding of the major public health problem of the dynamics of the burden of diabetes mellitus (DM). Most of the introductory part of the article deals with basic information on the pathogenesis of type I and type II diabetes mellitus, which would be unnecessary for the aim and readers of the study. At the same time, only a few sentences are devoted to the incidence and prevalence of the disease, which could be relevant to the study. A reader would like to know more about the justification of the aim of the study, taking into account the known and unknown information, the novelty of the study and the possible theoretical and practical application of the results.
In the part of Materials and Methods section, it would be advisable to include more information on the coding of death cases related to diabetes (all types). It is well known that most direct causes of death related to diabetes are complications of the disease, especially cardiovascular events in type II diabetes. Direct causes of the death (ketoacidosis, hypoglycaemia) are relatively rare. Is diabetes mellitus coded as direct cause of death or as a underlying principal diagnosis, or as an accompanying diagnosis? Also, a person can die with or because of diabetes – is a total DM mortality in a population (from any cause), in a DM population, or is it a cause-specific mortality? It is worth knowing more details about how the disease is coded in Poland using ISD codes, how the information flows to the registry. Is there a diabetes patient registry in Poland or in its regions, and how can the accuracy of underlying causes influence the accuracy of DM mortality statistics and its changes over time?
The calculation of measures clearly described and the use of joinpoint regression analysis and annual percentage change to describe dynamics is appropriate and clear. The only discussion that can be introduced is about average age at death, as averages depend on the distribution. Instead, medians are often used in studies.
The most important finding of the study is the increase of the DM SMR in Poland. In the Discussion this is mainly attributed to the increase in the incidence of DM. The data on the type II DM incidence are based on information from the Global Burden of Disease publication. However, the GDB takes data from the country specific reports. Are there publications on the DM incidence and prevalence and its change over time in Poland? The discussion indirectly links risk factors for the incidence with the mortality trends. However, incidence and prevalence depend on the accuracy of their estimation (reduction of unknown cases) and are not the only factor for the total and cause-specific mortality. The jointpoint regression analysis does not show a brak point for the SDR. The increase in the DM SDR during the Covid-19 pandemic is mentioned in the discussion, but not mentioned in the results (too short time period for regression?). The Standard Expected Years of Life Lost analysis shows some braking points. How can this be interpreted and explained? In addition, sex differences were found in all trends that need to be interpreted.
In Conclusions the empirical data in the study allow to agree, that DM in Poland is a burden for the healthcare, but the conclusion about the “adequacy” of the control depends on the definition of the term and its relative meaning. As the risk factors of the DM prevalence and control were not studied in this paper, indirect judgement of their impact on mortality is a kind of speculation – not direct evidence. It would be prudent to conclude from the observed increase in life expectancy in DM patients.
Minor points (for optional correction).
Line 16 and in other appropriate places. As the Polish population is an open population, a more scientifically appropriate measure for rates would be No over 100,000 person-years.
Line 30. “Civilisation metabolic diseases” is not a scientific term – there may be objections from social scientists who study civilisations.
Line 35. Probably the proportion is about the prevalence of DM. Reference?
Line 163. Full names of measures rather than abbreviations would be more appropriate in the titles of tables and figures.
Author Response
Dear Reviewer,
Thank you very much for your review and valuable suggestions. The changes introduced will undoubtedly improve the quality of the manuscript
The aim of the study is formulated as follows: “The aim of the study was to assess mortality and years of life lost from diabetes mellitus in Poland on a base of the register-based study between 2000 and 2022.” This assessment can certainly add to the knowledge and understanding of the major public health problem of the dynamics of the burden of diabetes mellitus (DM). Most of the introductory part of the article deals with basic information on the pathogenesis of type I and type II diabetes mellitus, which would be unnecessary for the aim and readers of the study. At the same time, only a few sentences are devoted to the incidence and prevalence of the disease, which could be relevant to the study. A reader would like to know more about the justification of the aim of the study, taking into account the known and unknown information, the novelty of the study and the possible theoretical and practical application of the results.
Justification, novelty and practical application of the study have been added in the Introduction section.
In the part of Materials and Methods section, it would be advisable to include more information on the coding of death cases related to diabetes (all types). It is well known that most direct causes of death related to diabetes are complications of the disease, especially cardiovascular events in type II diabetes. Direct causes of the death (ketoacidosis, hypoglycaemia) are relatively rare. Is diabetes mellitus coded as direct cause of death or as a underlying principal diagnosis, or as an accompanying diagnosis? Also, a person can die with or because of diabetes – is a total DM mortality in a population (from any cause), in a DM population, or is it a cause-specific mortality? It is worth knowing more details about how the disease is coded in Poland using ISD codes, how the information flows to the registry. Is there a diabetes patient registry in Poland or in its regions, and how can the accuracy of underlying causes influence the accuracy of DM mortality statistics and its changes over time?
In accordance with the law in force in Poland, the doctor confirming death fills out a death certificate in which he enters the initial, secondary and direct cause of death. The initial cause of death a disease, which was at the beginning of the morbid process and which caused the death, it may be also the injury or the poisoning as well as circumstances of the accident or the rape (external cause), which caused the death. A secondary cause of death is a disease, injury or circumstances of the accident, which were the consecution of the initial cause of death.The direct cause of death is defined as a disease, which was a definitive cause of the death as a consecution of the disease, injury or the poisoning as well as the circumstances of the accident, which were the initial and secondary cause of death.Statistical systems in most countries of the world, including Poland, use one of the causes of death entered in the card - the so-called the initial (primary) cause of death, because the most important thing is to isolate the disease giving rise to the pathological process that led to death. The procedure of coding causes of death in Poland has been described in Materials and methods and Limitations sections.
Information on diabetes registries in Poland has been added in the Introduction section.
The calculation of measures clearly described and the use of joinpoint regression analysis and annual percentage change to describe dynamics is appropriate and clear. The only discussion that can be introduced is about average age at death, as averages depend on the distribution. Instead, medians are often used in studies.
As suggested by the Reviewer, the mean age of death (with a 95% confidence interval) was changed to the median age of death (with a quartile range).
The most important finding of the study is the increase of the DM SMR in Poland. In the Discussion this is mainly attributed to the increase in the incidence of DM. The data on the type II DM incidence are based on information from the Global Burden of Disease publication. However, the GDB takes data from the country specific reports. Are there publications on the DM incidence and prevalence and its change over time in Poland? The discussion indirectly links risk factors for the incidence with the mortality trends. However, incidence and prevalence depend on the accuracy of their estimation (reduction of unknown cases) and are not the only factor for the total and cause-specific mortality. The jointpoint regression analysis does not show a brak point for the SDR. The increase in the DM SDR during the Covid-19 pandemic is mentioned in the discussion, but not mentioned in the results (too short time period for regression?). The Standard Expected Years of Life Lost analysis shows some braking points. How can this be interpreted and explained? In addition, sex differences were found in all trends that need to be interpreted.
The increase in DM SDR during the Covid-19 pandemic is clearly visible in Figures 1a and 1b, however, the relatively short duration of the pandemic did not result in a statistically significant change in trends. For this reason, these changes are not described in the Results section, but the increase in SDR is noted and commented on in the Discussion section. The Covid-19 pandemic could have resulted in a statistically significant increase in SEYLLp rates after 2018. This was probably related to the lower age of people who died from Type 1 diabetes mellitus in this period. Due to the relatively short time that has passed since the end of the pandemic, we believe that caution should be exercised when interpreting the changes that have occurred during this time.
In Conclusions the empirical data in the study allow to agree, that DM in Poland is a burden for the healthcare, but the conclusion about the “adequacy” of the control depends on the definition of the term and its relative meaning. As the risk factors of the DM prevalence and control were not studied in this paper, indirect judgement of their impact on mortality is a kind of speculation – not direct evidence. It would be prudent to conclude from the observed increase in life expectancy in DM patients.
Conclusions have been re-edited.
Minor points (for optional correction).
Line 16 and in other appropriate places. As the Polish population is an open population, a more scientifically appropriate measure for rates would be No over 100,000 person-years.
The mortality rate expressed as the number of deaths per 1,000 or 100,000 people allows us to assess the scale of deaths in relation to the size of the entire population. Data on the number of deaths and population are common all over the world, which allows the calculation of this mortality and compare between countries of all WHO regions.
Converting the mortality rate to a population provides information about the overall risk of death in the entire population over a given period of time. This gives a better idea of ​​how likely the average person in the population is to die, regardless of their age or duration of follow-up. Thanks to this, the mortality rate allows us to assess the health of the population as a whole and monitor changes over time (e.g. whether the mortality rate is increasing, decreasing or remaining stable).
The death rate per population is more intuitive for comparison than the rate per person-years because it directly shows how many people of the total population die in a given period of time.
Rates based on person-years are more appropriate in epidemiological studies that aim to accurately estimate the risk of death or disease over a specific period of time among people who are actually at risk. Rates per person-years are used when follow-up time varies for different people in a study, where we follow people from the time they are enrolled in the study until an event occurs (e.g., death) or the end of follow-up.
In demographic studies and public health analyses, it is more useful to calculate the mortality rate per total population. Therefore we would be grateful for the possibility of expressing the indicator per 100,000 population.
Line 30. “Civilisation metabolic diseases” is not a scientific term – there may be objections from social scientists who study civilisations.
It has been changed.
Line 35. Probably the proportion is about the prevalence of DM. Reference?
Please indicate precisely the sentence to which the comment concerns. As for the frequency of type 1 and type 2 diabetes, the reference is 3 and 4, respectively
Line 163. Full names of measures rather than abbreviations would be more appropriate in the titles of tables and figures.
It has been changed.

Reviewer 2 Report
Comments and Suggestions for Authors
Although the topic of the manuscript seems to be important, there are some parts which need to be corrected.
1. Abstract: Please write about the data that were used.
2. Abstract: Time trends were carried out?
3. Abstract:The meaning of SDR rate is uncertain.
4. Abstract: Is “Average percent change” a mistake for “Annual percent change”?
5. Abstract: Because SEYLLp means the ~ years per person, SEYLLp rates per 1,00000 persons is bizarre as an English phrase. In addition, the meaning of SEYLLp rates are uncertain. It is better to correct it also in the main text.
6. Conclusions in Abstract: The first sentence is rough.
7. Introduction: The authors did not mention about similar studies in the world or in Poland. Weren’t there any studies that investigated diabetes mortality trends in Poland?
8. Methods: What do SEYLLp and SEYLLd coefficients mean? Are “coefficients” necessary?
9. Table 1: How can we interpret the results of SEYLLd and SEYLLp? How are those indexes related to expected years of life lost?
10. Discussion: Weren’t there any limitations for this study?
Comments on the Quality of English LanguageSome parts need to be corrected.
Author Response
Dear Reviewer,
Thank you very much for your review and valuable suggestions. The changes introduced will undoubtedly improve the quality of the manuscript
Abstract:
- Please write about the data that were used.
Information about data has been added.
- Abstract: Time trends were carried out?
Yes, information is added in the Abstract.
- Abstract: The meaning of SDR rate is uncertain.
It has been clarified.
- Abstract: Is “Average percent change” a mistake for “Annual percent change”?
It has been corrected.
- Abstract: Because SEYLLp means the ~ years per person, SEYLLp rates per 100,000 persons is bizarre as an English phrase. In addition, the meaning of SEYLLp rates are uncertain. It is better to correct it also in the main text.
Formulas for the SEYLLp and SEYLLd have been added in the Material and Methods section.
Since the absolute number of years of life lost cannot be used for comparisons in long-term observations, due to the changing population size, it is recommended to calculate intensity indicators relating the absolute number of SEYLL to the number of inhabitants in the subsequent analyzed years. The resulting numerical values ​​of the quotients calculated in this way would be very small fractions, which could make interpretation difficult. Therefore, their values ​​are multiplied by a conversion factor, the value of which depends primarily on the frequency of occurrence of the phenomenon under study in the population. When analyzing mortality, a multiplier of 100,000 is usually used.
- Conclusions in Abstract: The first sentence is rough.
Conclusions have been re-edited in the Abstract and in the main text.
- Introduction: The authors did not mention about similar studies in the world or in Poland. Weren’t there any studies that investigated diabetes mortality trends in Poland?
Information on diabetes registries in Poland has been added in the Introduction section. Results from studies on mortality have been added in Discussion section. Simultaneously, it is worth noting that our study is the first to focus on evaluating the health burden of diabetes in the Polish population, using comprehensive data on mortality and years of life lost due to diabetes over a span of more than 20 years. Therefore, a direct reference to similar studies is not possible. There is, however, a reference to a study covering mortality trends in European countries
- Methods: What do SEYLLp and SEYLLd coefficients mean? Are “coefficients” necessary?
"coefficients" for SEYLLp and SEYLLd hav ebeen removed
- Table 1: How can we interpret the results of SEYLLd and SEYLLp? How are those indexes related to expected years of life lost?
The interpretation of SEYLLd and SEYLLp is described in the Material and methods section.
The more deaths due to diabetes mellitus per 100,000 people in the study population, the greater the number of years of life lost and the higher the SEYLLp value. Standard expected years of life lost draw attention to the social and economic aspect of losses related to premature mortality.
The decreasing SEYLLd value means a lower average number of years of life lost by each person dying due to diabetes mellitus, which may indicate a later age of onset of the disease in the deceased people or an extension of their life span through early detection and effective treatment of the disease.
- Discussion: Weren’t there any limitations for this study?
Limitations section has been added.

Round 2
Reviewer 1 Report
Comments and Suggestions for Authors
Thank you very much for careful and complete answers and corrections. The article is nearly finished, just:
· In Figure 1 C (SEYLLp), the jointpoint regression line shows us changes (break points) for women in 2004 and for men in 2015. Is there any explanation for that?
· The section “Limitations” deals more with methods (data sources) than limitations and can be transferred to “Materials and Methods”. Moreover, this information is already partly there.
· I completely agree with your arguments regarding rates per population and not per person-time (common tradition, more intuitive, etc.). Just some (American leading) schools are insisting that rate never can be over population, but time in contrast to cumulative proportions in closed populations. Anyway, person-years are acquired by the average population in a time period. But I do not insist on turning to person-years, and you can leave as you like.
· Sorry for my remark about line 34 - misinterpretation. Everything is good.
Author Response
Dear Reviewer, thank You very much for all suggestions.
Figure 1 C (SEYLLp), the jointpoint regression line shows us changes (break points) for women in 2004 and for men in 2015. Is there any explanation for that?
A sentence has been added to the Discussion section relating to the observed result.
The section “Limitations” deals more with methods (data sources) than limitations and can be transferred to “Materials and Methods”. Moreover, this information is already partly there.
Limitations section has been re-worded and transferred to the Material and methods section.
I completely agree with your arguments regarding rates per population and not per person-time (common tradition, more intuitive, etc.). Just some (American leading) schools are insisting that rate never can be over population, but time in contrast to cumulative proportions in closed populations. Anyway, person-years are acquired by the average population in a time period. But I do not insist on turning to person-years, and you can leave as you like.
Thank you very much. If it is possible we leave it in the previous form.
Sorry for my remark about line 34 - misinterpretation. Everything is good.
Thank you very much for the clarification.
